# Study protocol for a systems evaluation of an infant mental health service: Integration of 'Little Minds Matter' into the early years system

Sarah Masefield[1], Alison Ellwood[2], Sarah Blower[1]*, Josie Dickerson[2], Rachael H. Moss[2], Sara M. Ahern[2]

1 Department of Health Sciences, University of York, York, North Yorkshire, United Kingdom, 2 Born in Bradford, Bradford Teaching Hospitals National Health Service Foundation Trust, Bradford, West Yorkshire, United Kingdom

* sarah.blower@york.ac.uk

## Abstract

Infant Mental Health relates to how well a child develops socially and emotionally from birth to age three. There is a well-established link between parent-infant relationship quality, Infant Mental Health, and longer-term social and emotional development there is a lack of evidence-based interventions that support the parent-infant relationship and/or protect against poor Infant Mental Health. Little Minds Matter is a specialist Infant Mental Health service developed in Bradford (UK) to support parent-infant relationships by providing training and consultation for professionals and direct clinical work to families. The successful implementation of this intervention depends upon how well it becomes embedded within, or integrated into, the early years system. For the purposes of this study, the early years system includes health, social and education services that support child health and development from conception to primary school entry at age five. This study protocol aims to apply a systems approach to evaluate this service and provide a perspective on the process of embedding a complex service within a healthcare system. Multiple methods will be used to investigate embeddedness within the wider early years system. Routinely collected quantitative data about the service will be used to develop a system map showing interaction with related services. Qualitative data will be collected at two time points through interviews with individuals involved in the design and provision of the service, and professionals working within the early years system. Framework analysis will be used to analyse the data inductively and deductively within a systems approach. The findings from this study will provide evidence to inform the ongoing implementation of the service for providers and commissioning bodies. Exploring the application of a systems approach in this clinical context will have application more broadly for researchers evaluating complex interventions and services within a wider system.

**Data Availability Statement:** Data from this study can not be made available due to the qualitative

nature of the study and ethical limitations. However, deidentified research data from the BiB and BiBBs cohorts can be made available to those who agree to the terms and conditions of this. Readers who wish to request data from the CiB or BiBBS cohorts should contact this email BorninBradford@bthft.nhs.uk This contact will be stable and these requests can be then forwarded on to the most appropriate person for management.

**Funding:** This research is funded by two awards. The National Lottery Community Fund (previously the Big Lottery Fund) as part of the A Better Start programme (Ref 10094849). Awarded to JD, SA, SB NIHR Yorkshire and Humber Applied Research Collaboration (ARC-YH; Ref: NIHR200166). Awarded to JD, SB No commercial companies funded the study or provided a salary to the authors of this paper. The study sponsor is Bradford Teaching Hospital Foundation Trust Research Management & Support Office - Bradford Institute for Health Research (bradfordresearch.nhs.uk) The funders did not and will not have a role in study design, data collection and analysis, decision to publish, or preparation of the manuscript.

**Competing interests:** The authors have declared that no competing interests exist.

# Introduction

Infant mental health (IMH) relates to how well a child develops socially and emotionally from birth to age three [1]. Secure parent-infant relationships are the foundation for protecting IMH [2]. If an infant feels safe and secure, and experiences the world as a consistent, loving place, they are more likely to build a healthy brain and grow into a confident person who enjoys happy relationships [3]. The association between parent-infant relationship quality, IMH, and children's longer term social and emotional development is well evidenced [2, 4–6]. Despite this, there is a lack of evidence-based interventions that support the parent-infant relationship [3, 7] and/or protect against poor IMH [8].

In their recent evidence report, the Parent-Infant Foundation identified a 'baby blind-spot' within current Child and Adolescent Mental Health Service (CAMHS) provision in England, with many services not accepting referrals for children under 24 months of age, thereby missing opportunities to intervene at the earliest opportunity [9]. The Parent-Infant Foundation consensus statement recommends improving outcomes in the first 1,001 days of a child's life, establishing specialist IMH services at a local level and providing care continuity with a skilled workforce [10]. There are currently 45 such specialist parent-infant relationship or IMH teams in the UK offering theoretically grounded interventions in their therapeutic provision [11]. The Parent-Infant Foundation provide guidelines on the design and development of IMH services in line with the existing evidence of effectiveness [12]. To provide support to those families in need, these services need appropriate referrals from professionals working across the early years services who come into contact with families. In this study we define the early years system as consisting of all the health, social and education services that support child health and development from conception to primary school entry at age five. However, identification of issues in the parent-infant relationships by these professionals may be challenging. Whilst National Institute for Health and Care Excellence (NICE) guidance highlights the importance of assessing the relationship parents have with their children, there are no assessment tools recommended for use in the critical first 12 months after birth, and those recommended for pre-school children require clinical expertise to complete [13]. Consequently, healthcare professionals working in universal settings with families with young children may lack the confidence and/or ability to assess this relationship [14, 15].

To address these challenges, one such specialist IMH service, Little Minds Matter (LMM), has been designed to provide direct therapeutic support for the parent-infant relationship at any time from pregnancy up to 24 months post birth. LMM has been providing services to families and professionals since April 2018. Prior to this mental health care provision to children in Bradford was often restricted to those above 24 months of age and professionals described a lack of confidence in supporting families with concerning parent-infant relationships. The aim of the service is to influence the wider early years system to raise awareness of IMH in the community and empower professionals to confidently identify issues and support parents within their usual practice. Success of the intervention depends on becoming embedded within the early years system, so that relevant services have an awareness of IMH and are able to utilise the services offered by LMM. By this we mean that relevant projects and services within the system have an awareness of IMH and the services offered by LMM and have support and opportunities at both an organisational and systems level to interact with the service. To explore the concept of embeddedness, or integration, of the service into a wider system, we will look at both how the service and the system have changed over time and the dynamic relationship of these changes. A useful way to conceptualise and examine this influence is by understanding both the extensiveness of the intervention across the system (how far has it reached) and the intensiveness of its integration into routine practice [16]. This will involve

exploring both the pre-intervention context as well as the current context in which the service functions.

In 2021 the Medical Research Council updated their guidance for evaluating complex interventions to emphasise the value of understanding the context within which an intervention operates, thus enabling a richer comprehension of the mechanisms required to bring about change [17]. This requires exploration of integration within the existing system and the impact of the intervention upon existing activity, displacement of resources and ongoing relationships across and within the system. In accord with this updated guidance, the aim of this protocol is to describe the process evaluation of the LMM intervention using a systems evaluation approach. Our five objectives are:

1. to describe what the early years system looked like before LMM was implemented (i.e., the pre-intervention context).

2. to explore changes that have occurred in the early years system since LMM was first introduced.

3. to identify and explore the ways in which LMM has become embedded within (i.e., coupled with) the early years system.

4. to explore factors that have facilitated or hindered the implementation of LMM.

5. to explore whether LMM has succeeded in increasing the knowledge and understanding of IMH in professionals working within the early years system.

## Methods

### Design

This is a systems evaluation approach using multi-methods including quantitative and qualitative data and systems mapping.

### Study setting

This study is being conducted in Bradford, a large district in the North of England with a young, ethnically diverse population, many of whom live in areas of high deprivation. The early years system is defined as all the services that work with families with young children to promote children's health and development from conception to primary school entry at age 5. This includes universal care (e.g., midwifery and health visiting), targeted services (e.g., mental health services and social care) and a wide range of voluntary and charitable sector (VCS) organisations delivering support and interventions. Significant to the early years services in Bradford are two large programmes of work providing additional support to the early years system in inner-city areas: Better Start Bradford, a National Lottery Community Fund programme that commissions a number of preventative interventions to improve socio-emotional development, language and communication development and nutrition in children aged 0–3 [18]; and the Reducing Inequalities in Communities programme that funds interventions to reduce inequalities across the life course including the early years [19]. Evaluation of the implementation of these projects was considered important, therefore a team of dedicated researchers, the Better Start Bradford Innovation Hub, have worked alongside projects, routinely monitoring delivery and providing ongoing feedback [18].

### The intervention

Little Minds Matter is a specialist IMH service developed to support the parent-infant relationship during pregnancy until the infant is 24 months old. We acknowledge that not all

caregivers are parents in a biological or legal sense, for clarity we also include within the scope of this paper parents as those who provide daily care to a child, and act as a 'parent' whether they are biological parents or not [20]. The service is commissioned by both the Better Start Bradford and Reducing Inequalities in Communities programmes and is delivered by Bradford District Care NHS Foundation Trust. According to the theory of change for LMM which was produced during the service design process, a key mechanism for an improved parent-infant relationship is *mind-mindedness* which is defined as the ability of parents to tune in to their children's thoughts and feelings [21]. The service acts on this mechanism in a number of ways. Firstly, by raising awareness across the early years workforce and wider community of the importance of the parent-infant relationship to later child outcomes. Secondly, by training and supporting the early years workforce who work directly with families to identify and support families with 'concerning' relationships. Thirdly, by providing direct clinical work with families in need. The service defines a 'concerning' relationship as one where a parent does not appear to be aware of their infant's feelings and are unable to exhibit mind-mindedness. Practitioners' ability to identify concerning relationships can be limited, therefore issues of mind-mindedness and poor IMH are likely to be underdiagnosed. Through four strands of activity LMM seeks to raise awareness of the importance of parent-infant relationships and IMH with practitioners and improve access to support for families (see Fig 1: Overview of Little Minds Matter service) to enact change and improve IMH in Bradford. Strand one is delivered by a VCS partner and strands two, three and four are delivered by a team of approximately 10 professionals. This includes Clinical Psychologists, Family Therapists and those with a

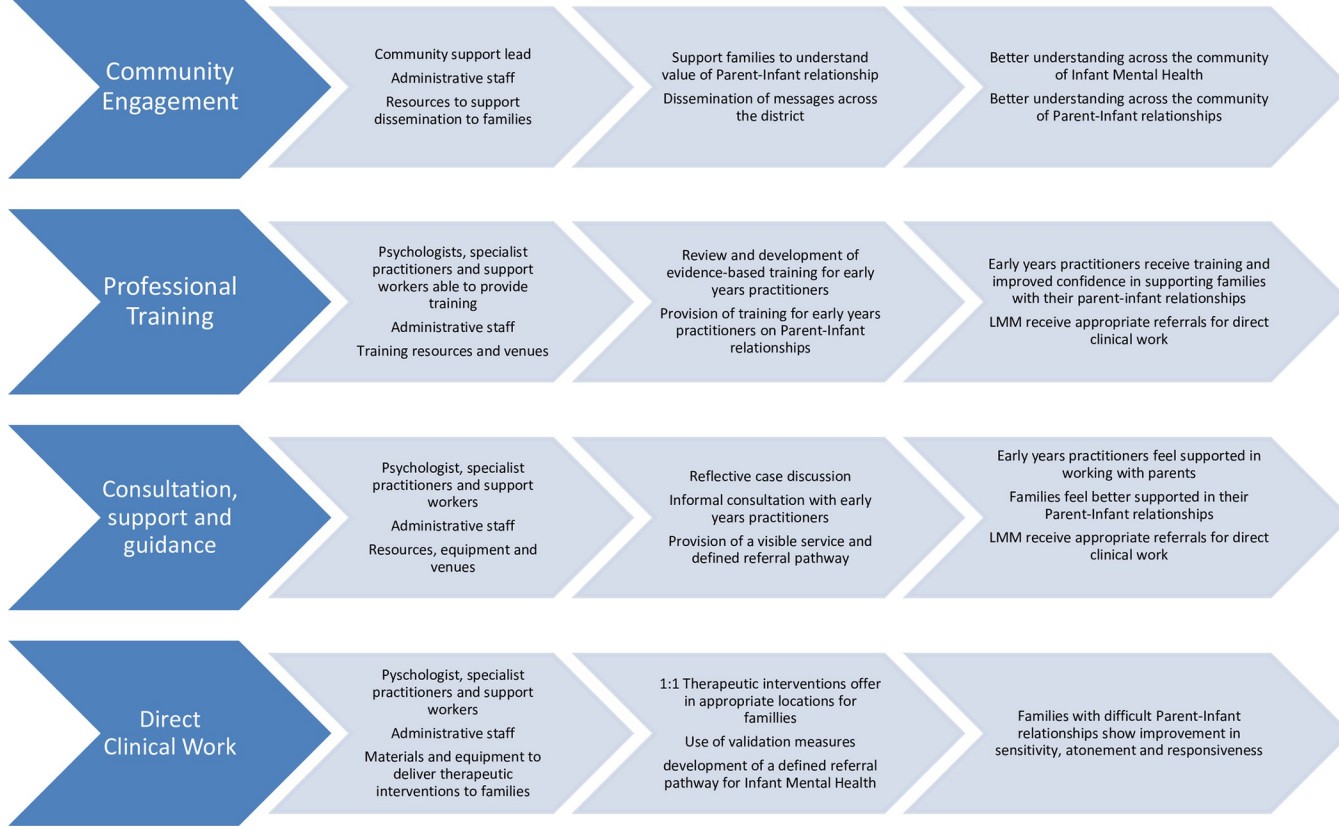

**Fig 1. Overview of Little Minds Matter service.**

background in Health Visiting, Midwifery and Social Care settings. This service can be accessed by professionals in the early years system for advice and support and to refer families into the service for therapeutic work. Professionals can access the service by telephone and email, and families can self-refer to the service through a website or telephone contact. The service has undergone a thorough service design process, including development of a theory of change, logic model and agreed implementation targets for each strand for which routine monitoring data is gathered. Further information relating to this design process can be requested from the corresponding author.

## A systems approach to evaluation

The 2021 Medical Research Council framework for evaluating complex interventions includes the context within which an intervention operates with specific consideration of the conditions needed to realise the mechanisms for change [17]. In a systems approach to evaluating health interventions, interventions are conceptualised as 'events in systems that either leave a lasting footprint or wash out depending on how well the dynamic properties of the system are harnessed' [16]. An essential initial component of a systems approach is exploring and understanding the context into which the intervention is introduced and with which it interacts, essentially the pre-intervention context. Additionally, mapping of the system of interest is often a preliminary and integral part of a systems evaluation. The pre-intervention context and system map are then used as key tools in the systems approach to enable four avenues of inquiry: 1) how the intervention couples with the context; 2) tracking changes in relationships within the system; 3) the distribution and transformation of existing resources; and 4) the displacement of pre-intervention activities [16]. Using this approach, our study will look at how the intervention and the early years system have changed over time in relation to IMH, and the dynamic relationship of these changes. In our study, change over time is explored through data collection at two different time points 9–12 months apart (see objectives outlined in Table 1). Additionally, we have limited capacity to assess the redistribution and transformation of existing resources as we are not accessing financial records as part of this study.

## Development of a system map

To describe the early years system in Bradford and identify key stakeholders to participate in interviews we will conduct a scoping exercise to identify early years services and projects in Bradford. This scoping exercise will involve exploring existing pathways within the early years system (between these services and LMM) and conversations with stakeholders including health and social care professionals, researchers and those involved in the LMM service design process. From this we will produce a list of services and projects that work with families in the early years system to promote children's health and development. These services and projects will then be placed into a systems map, with services and projects placed in concentric circles, based upon the degree to which supporting the parent-infant relationship is a core objective of that service's remit (see example map in Fig 2). This process will be iterative and based on the research team's extensive understanding of the existing early years system in Bradford and LMM service design documents, which describe the expectations of interaction between LMM and other services. Implementation data is routinely collected about which organisations and professional groups have interacted with LMM via the training, consultation and clinical work strands since the service's launch. This quantitative data will then be plotted onto the system map to provide a visual description of the way in which services and professionals have engaged with LMM.

**Table 1. Objectives and data processes.**

| Objective | Methodological approach | Participants or Data source | Time point | Data collection method | Data analysis method |
|---|---|---|---|---|---|
| To describe what the early years system looked like before LMM was implemented (i.e., the pre-intervention context). | Qualitative methodology | Individuals involved in the development of LMM (group one); individuals employed in roles supporting families with young children (group three) (see Fig 2) | Time one | Interviews and Focus Groups | Framework analysis exploring patterns in the data across groups at time one |
| To explore changes that have occurred in the early years system since LMM was first introduced (to include the displacement of any pre-intervention activities and transformation of existing resources). | Qualitative methodology | Individuals involved in the development of LMM (group one); Individuals employed by LMM but not involved in service design (group two); individuals employed in roles supporting families with young children (group three) | Time one | Interviews and Focus Groups | Framework analysis exploring patterns in the data across groups at time one and time two 12 months later |
| To identify and explore the ways in which LMM has become embedded within (i.e., coupled with) the early years system. | Quantitative Methodology | Routinely collected data relating to training, consultation and referrals | Time one and time two | Routinely collected count data | plotted onto a system map |
| | Qualitative methodology | Individuals employed by LMM but not involved in service design (group two); individuals employed in roles supporting families with young children (group 3) | | Interviews and Focus Groups | Framework analysis exploring patterns in the data across groups at time one and time two 12 months later |
| To explore factors that have facilitated or hindered the implementation of LMM. | Qualitative methodology | Individuals employed by LMM but not involved in service design (group two); individuals employed in roles supporting families with young children (group 3) | Time one and time two | Interviews and Focus Groups | Framework analysis exploring patterns in the data across groups at time one and time two 12 months later |
| To explore whether LMM has succeeded in increasing the knowledge and understanding of IMH in professionals working within the early years system. | Qualitative methodology | Individuals employed in roles supporting families with young children (group 3) | Time one and time two | Interviews and Focus Groups | Framework analysis exploring patterns in the data across groups at time one and time two 12 months later |

## Qualitative data collection

The systems map will then be used within qualitative interviews to help elucidate the pre-intervention context, the fit of LMM within the system and the extent of interaction between services and LMM. It will also facilitate other lines of questioning, including developing an understanding of any unintended consequences of introducing the LMM service into the existing early years system. Qualitative data collection will be undertaken with three different groups of stakeholders: group one, individuals involved in the development of LMM; group two, individuals employed by LMM but not involved in service design; group three, individuals employed in roles supporting families with young children (see Table 1 detailing objectives and data processes). Data will be obtained using interviews and focus groups as appropriate. For groups two and three, interviews will be undertaken at two time points to observe change in the system over time. Interviews will take place face-to-face or through online video conferencing depending on the participant's choice. To report our qualitative findings, we will adhere to the Standards for Reporting Qualitative Research (SRQR) and the COnsolidated criteria for REporting Qualitative research (COREQ) [22].

## Participants

Study participants are eligible for interview if they are an individual working within the early years system, who is either:

- Employed by the National Health Service (NHS), social care, Better Start Bradford and related organisations/projects, local authority, and VCS in a service which supports the

VCS Voluntary and
Charitable Sector service
P Private service
S Statutory service
BSB Better Start Bradford
service

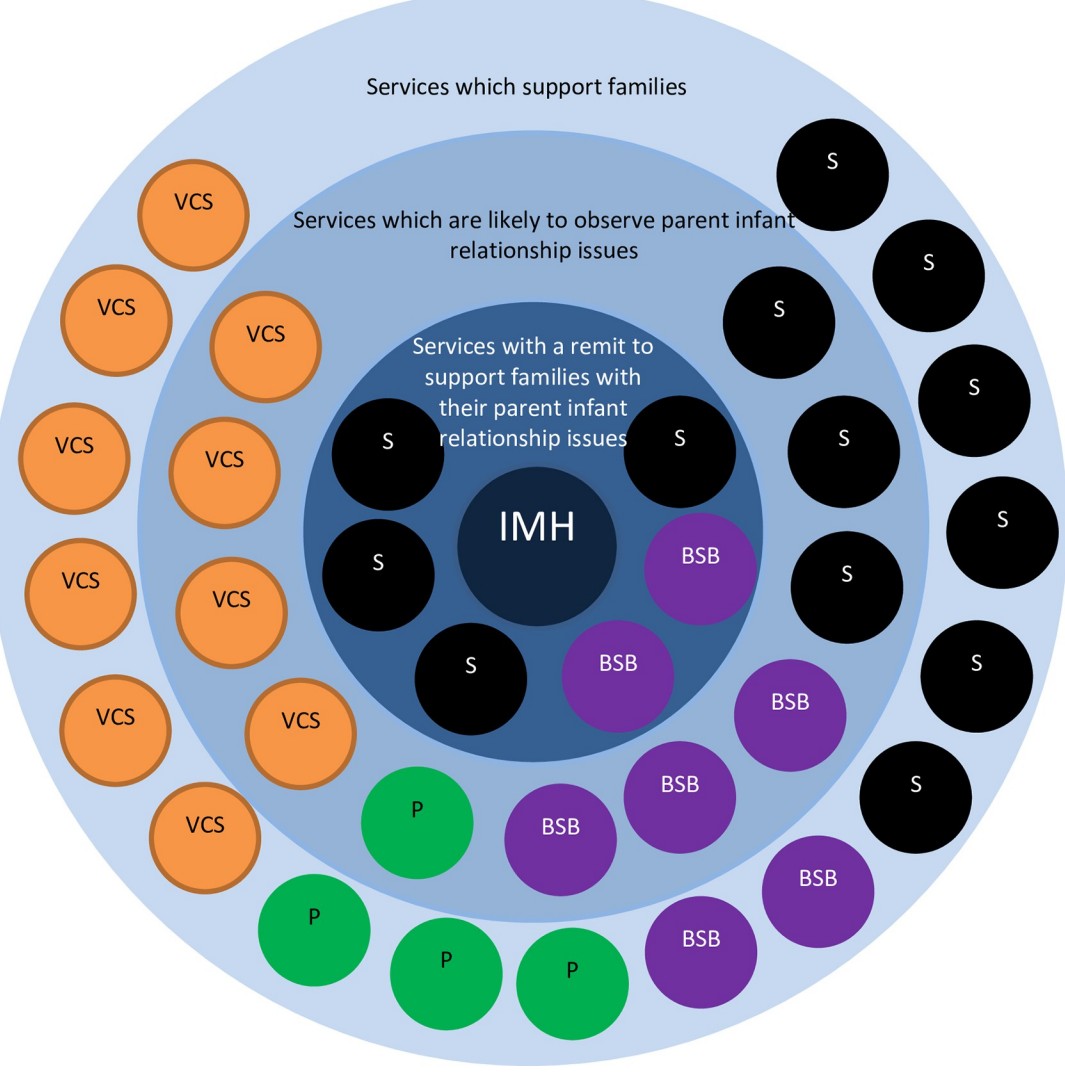

**Fig 2. Example template map.**

parent-infant relationship with sufficient oversight of the project to understand engagement with the LMM service, such as team-leaders or managers.

OR

- Were involved in the design and development of the LMM service.

We aim to recruit between 25–35 participants at each time point: four in group one; approximately seven in group two and 25 in group three. For group one and two these numbers are based upon the number of individuals involved in the design and current provision of LMM. Group one participants are identified by liaising with the Better Start Bradford team and group two participants will be identified by the clinical lead of LMM. Recruitment to group three will be dependent on the services identified in the systems map and interviews with participants in group one and group two. Initial scoping suggests that around 25 services may be included on the Bradford early years system map. This number may be revised and will depend upon the agreement and availability of invited individuals to participate.

Sampling will be purposive, targeting staff working at the team lead or managerial level in organisations that are considered most appropriate to gain organisational oversight and a wider view of service integration and training within the team. In most cases only one person per service will be interviewed and interviews with participants from services with a remit to support the parent-infant relationship will be prioritised (i.e., those in the inner circles of the systems map, see Fig 2). This approach will support data saturation given the range of participation from varied projects.

The study participants will be approached by email from the primary researchers (SM and AE). Email addresses will be accessed through contacts with the Better Start Bradford and LMM teams as well as contacts within the Better Start Bradford Innovation Hub following agreement by the relevant research and development offices. Emails will include the consent form and study information sheet. The information sheet includes details of the full study, indicating which participant groups we intend to approach for a second interview. Written or verbal consent will be taken prior to interview. Participation can be face-to-face or through video call therefore verbal consent will be recorded separately from the interview and stored in a different location on the Bradford Teaching Hospital Foundation Trust's online secure archive in password protected folders.

## Data collection

Data collection will be scheduled to ensure that those involved in the design and commissioning of LMM and LMM staff are interviewed first. This will allow for review of the system map (with any significant omissions added or clarifications made prior to the other interviews) and review of the list of organisations that the researchers intend to invite to interview in group three. Additionally, information will be captured and review of the topic guides for the subsequent interviews with staff from other services in the early years system will be undertaken. At times it may be more viable to collect data through focus groups rather than one-to-one interviews. This decision will be taken pragmatically on a case-by-case basis according to the preference of the participants. At time point one, all three groups will be questioned on a range of subjects as described in Table 1, including their role in the service, understanding of the LMM service, the reach of LMM across the early years system and interaction between LMM, other services and professional groups. For groups one and two, a copy of the draft map will be shared with participants prior to their data collection session. This will enable these participants to review the map in preparation for questions in the interview about which services have existing relationships with LMM, which additional services it would be desirable to establish relationships with and how IMH provision may overlap between different services. Group three will not be asked to review the map as at this time as we are primarily interested in their relationship with LMM and not with other services in the early years system. Group three participants will be questioned at time one and two about their understanding of IMH and how they feel this has changed over the duration of the project. Interviews are expected to last 1–1.5 hours for groups 1 and 2 and 30–45 minutes for group 3, it is hoped that shorter interview times will facilitate involvement from busy individuals.

At time point two, 9–12 months later, participants from groups two and three will be invited to attend a second interview or focus group. Wherever possible this will be the same individuals, agreement to approach participants again will be discussed at the end of their time one interview, but where these people have moved on an alternative appropriate individual will be approached. Again, they will be asked questions about their service, understanding of the LMM service, the reach of LMM into and interaction between LMM, their/other services and professional groups. At this time point group two will also review the map again, which

will now include additional data from the time between qualitative data collection periods. The review of the map and interview questions will facilitate exploration of change in the early years system over the intervening period. Separate topic guides have been created for the different groups and will be used by the member of the research team conducting each interview. The topic guides may be amended following interviews with participants in group 1 and 2. With this approach we will achieve a whole-system perspective, including an understanding of the pre-intervention context.

## Ethical approval and study governance

Health Research Authority (HRA) approval was granted on 25th August 2022 (IRAS project ID: 311645 REC reference: 22/HRA/3477) for research undertaken within the NHS Trusts and Sponsorship approval received for Bradford Teaching Hospitals NHS Foundation Trust on 3rd October 2022 and for Bradford District Care NHS Foundation Trust on 18th October 2022. University of York Department of Health Sciences Research Governance Committee approval was granted for research undertaken with non-NHS trusts via a Chair's Action letter (30 September 2022).

## Data management

Data will be managed confidentially in line with ethical approval. Interviews will be transcribed and anonymised then the recordings deleted. All data related to the map development, transcripts and consent information will be stored in the Bradford Teaching Hospital Foundation Trust online secure archive in password protected folders. Transcripts will be stored separately from identifiable and consent information. Transcripts will be managed on Computer Assisted Qualitative Data Analysis Software (CAQDAS) software, NVivo version 14 (Lumivero, 2020) [23]. Transcript data and consents will be stored for ten years after which time they will be destroyed.

## Data analysis

**Quantitative data.** Descriptive statistics will provide counts of the number of professionals and organisations within the early years system who have engaged with each of the strands of the LMM service.

**Qualitative data.** Framework analysis will be used to analyse the data using Computer Assisted Qualitative Data Analysis Software (CAQDAS) software, NVivo. Framework Analysis [24] will be used to organise and analyse the data. This approach is well suited to this study because it is anticipated that a large amount of data will be generated, requiring a structure to manage and reduce the data to respond to the research questions. It uses a matrix format to organise the data into rows or cases (representing participants) and columns (representing codes) to support comparison of data across, as well as within cases [24].

Verbatim transcription will be completed by a transcription service approved by Bradford Institute for Health Research. Where focus groups are undertaken, we will approach the data as a group interview transcript analysed in the same way as the individual interview transcripts. Once transcribed, familiarisation with the data will be increased by reading of the transcripts while listening to the recorded interview. Once familiarised with the data three transcripts will be selected, one from each participant group. Using these three transcripts a coding framework will be developed. Initially codes will be deductively informed from the existing literature, in addition, two researchers will engage in open coding to enable themes to emerge from these three interview transcripts. Therefore, the coding will be both inductively and deductively informed, limiting the extent to which key findings might be missed. Once

developed and agreed the framework will be utilised to code the remaining transcripts. As with the coding process, themes will be both inductively and deductively derived through discussion with the wider research team. We will not employ restrictions on coding, such as stopping analysis in response to data saturation because of the variety of professionals and services interviewed, new themes and relevant information may arise in each interview. This process will reduce the potential for bias and enhance the quality of the analysis.

## Discussion

This study will use a multi-methods systems approach, integrating implementation data, systems mapping and qualitative data to evaluate the extent and ways in which the LMM service has become embedded within the early years system in Bradford (UK) to improve IMH. The outlined study will make an important contribution to knowledge.

Systems approaches, a defining feature of this study, are relatively new to health research and in keeping with the most recent guidance from the Medical Research Council [17]. The integration of a service into a wider system is often a key factor in the success of the service and the impact it has across the system. The approach outlined in this protocol offers a way to evaluate whether a service has become embedded and is able to influence a system. Such approaches may offer a pragmatic solution to evaluation, given there are often fluctuations in service provision and time limitations upon service providers and researchers. Furthermore, this study will provide future researchers with a valuable example of conducting such an evaluation of a service. The study findings will provide learning for the LMM service and early years commissioners in Bradford to enable them to adapt their service as required. Moreover, findings will provide a comprehensive understanding of the barriers and successes to system change for other similar services to review their practice. The depth of this study will also provide an understanding of the potential for the unintended consequences of implementing a new service into an existing system.

This study will only describe the impact of one service on one system in one location, however the knowledge and understanding gained about IMH services will provide an evidence base for similar service development elsewhere. In particular, this study has particular relevance to the current UK government initiative, the Best Start for Life [4], given the focus this policy has on the first 24 months of life and developing strong parent-infant relationships. A further limitation is that the understanding of the pre-intervention context is being accessed retrospectively by asking participants to reflect on provision before the introduction of LMM. This may impact on the understanding that can be gained about how much change has been brought about by the LMM service into the early years system in Bradford. However, by interviewing people involved in the service design stage we will gain an insight into the pre-intervention and through this approach we are developing the systems approach method and demonstrating how it can be adapted for existing services where it is desirable to explore system change over time.

## Conclusion

IMH services appear to be a valuable early years service, but their successful implementation depends upon becoming embedded within the wider early years system. Evaluation of this with a system approach will provide learning for these services on the challenges and successes of embedding a complex service within a healthcare system and make an applied contribution to the understanding of systems approaches.

## Author Contributions

**Conceptualization:** Sarah Masefield, Sarah Blower, Josie Dickerson, Sara M. Ahern.

**Funding acquisition:** Josie Dickerson, Sara M. Ahern.

**Investigation:** Sarah Masefield, Alison Ellwood, Rachael H. Moss.

**Methodology:** Sarah Masefield, Alison Ellwood, Sarah Blower, Josie Dickerson, Sara M. Ahern.

**Project administration:** Alison Ellwood.

**Supervision:** Sarah Blower, Josie Dickerson.

**Writing – original draft:** Sarah Masefield, Alison Ellwood, Sarah Blower, Josie Dickerson.

**Writing – review & editing:** Sarah Masefield, Alison Ellwood, Sarah Blower, Josie Dickerson, Rachael H. Moss, Sara M. Ahern.

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
