## [Decision Letter · Decision Letter 0]

19 Jul 2023

PONE-D-23-15381Study protocol for a systems evaluation of an infant mental health service: integration of ‘Little Minds Matter’ into the early years system.PLOS ONE

Dear Dr. Ellwood,

Thank you for submitting your manuscript to PLOS ONE. After careful consideration, we feel that it has merit but does not fully meet PLOS ONE’s publication criteria as it currently stands. Therefore, we invite you to submit a revised version of the manuscript that addresses the points raised during the review process.

We look forward to receiving your revised manuscript.

Kind regards,

Veincent Christian Pepito

Academic Editor

PLOS ONE

Additional Editor Comments:

Dear Author, thanks for your submission. It looks promising; however, I would like to clarify a few things:

1. I want you to clarify what specific type of "evaluation" you are carrying out. Is it an impact evaluation? Is it a process evaluation? This is to manage expectations for the reader.

2. I want you to give more details about what the intervention is by actually showing its theory of change (i.e., from inputs to process, to output, to outcome, and to impact), instead of just its parts. It would also be helpful what specific parts or aspects of the ToC will you be actually evaluating (which will answer Comment 1).

3. One of the objectives is to assess whether LMM increased knowledge and understanding of IMH (Line 144-145). However, I do not actually see what specific methodology you will use to answer this objective and how you will attribute any change in knowledge to the LMM program instead of other externalities.

4. On a more general note, I would appreciate it if you could specify how you would answer each of the objectives you have listed in Lines 134-145. Who are the respondents, what is the evaluation design used, what frameworks should be used, what analysis methods should be used, etc. I think you will also be using an implementation research framework to answer the facilitators and barriers objective, I surmise?

5. It is exemplary that you want to document how the embedding of the LMM affected the status quo. I want you to scale this up by also describing any assessments of unintended consequences that you would be doing on top of what has been described.

Minor comments:

1. Do not confuse efficacy with effectiveness.

2. Fix referencing.

Please also see the comments of the reviewers and see to it that these are addressed so that we can publish your protocol soon. Thank you.

Reviewers' comments:

Reviewer's Responses to Questions

**Comments to the Author**

1. Does the manuscript provide a valid rationale for the proposed study, with clearly identified and justified research questions?

Reviewer #1: Partly

Reviewer #2: Yes

2. Is the protocol technically sound and planned in a manner that will lead to a meaningful outcome and allow testing the stated hypotheses?

Reviewer #1: Partly

Reviewer #2: Partly

3. Is the methodology feasible and described in sufficient detail to allow the work to be replicable?

Reviewer #1: No

Reviewer #2: Yes

4. Have the authors described where all data underlying the findings will be made available when the study is complete?

Reviewer #1: Yes

Reviewer #2: Yes

5. Is the manuscript presented in an intelligible fashion and written in standard English?

Reviewer #1: No

Reviewer #2: Yes

6. Review Comments to the Author

You may also provide optional suggestions and comments to authors that they might find helpful in planning their study.

Reviewer #1: Here are some of my comments and questions related to the manuscript:

Introduction:

- The aim of the study states: “To describe the evaluation of LMM intervention using a systems evaluation approach”. Are there specific objectives to the study? In line 131, the authors have explained that it will look into how the system has changed over time and the dynamic relationship of the changes. Lines 136 onwards also lists study objectives. Clarify the general and specific objectives of the study and include these in the introduction.

- The introduction describes the significance of the infant mental health and the program. Since the main objective of the proposal is to evaluate, improve this section with additional literature on the systems based approach to evaluation to help readers better understand the purpose of the protocol.

Methodology:

- Provide an illustration of the conceptual framework of the systems approach to evaluation

- It was difficult to follow what the research group’s methodology is for the data collection. Could this be better supported through a straightforward summary or perhaps illustrated through a diagram? Also the differentiation of the participant groups through a table that summarizes the data to be obtained, data collection tool to be used, type of analysis, etc.

- Outline potential assumptions and describe process more systematically to allow readers, policy-makers and other researchers to understand and even replicate the proposed approach to evaluation

Study setting and The intervention

- Since this is a program that is already running, how long has it been running?

- How is the service availed within the study setting?

Development of a system map

- What is the basis of the system map? Is there a guiding framework being followed particularly the 3 categories for eh core objectives that services remit?

Participants

- Clarify the inclusion criteria (Line 177). Line 189 describes targeting staff working as team lead or in managerial levels. This and other inclusion criteria may be most helpful in understanding the process if it were included in the eligibility list describing the participants.

- The number of respondents for the groups are different. Describe the basis of the sampling.

Data collection / analysis

- Since the group will not be stopping based on data saturation, how will the research team ensure that adequate information has already been gathered?

- The limitation on the pre-intervention context should be supported by secondary literature review.

Conclusion states that the study will evaluate whether a services has become embedded and influences a system. Which areas of the protocol will be answering this objective?

Improve language and grammar of the manuscript to provide a clearer and cohesive narration of the project.

Reviewer #2: - Congratulations on the conceptualization and implementation of the Little Minds Matter Program!! Early childhood matters!

- Before embedding the Little Minds Matter in early years system, its validity as an effective intervention in improving infant mental has to be established. Please provide more information on this, i.e. initial outcomes studies on improvement of parent-infant relationships, etc. after completing the program.

- Line 169. Would it be possible to include recipients of the program as another stakeholder group as it is noted that those interviewed are all service providers

- Line 222. On the contrary, wouldn't it be beneficial for Group 3 to review the service map to see how LMM relates to the other services in the early years system to decrease the chances of it duplicating existing services and also to see how it can work synergistically with other programs?

- Line 295. I am not clear about what your parameters are about LMM being "embedded" and "influencing" the system

7. PLOS authors have the option to publish the peer review history of their article (what does this mean?). If published, this will include your full peer review and any attached files.

Reviewer #1: **Yes: **Angel Belle Dy

Reviewer #2: No

---

## [Author Response · Author response to Decision Letter 0]

7 Sep 2023

Bradford Institute for Health Research

Temple Bank House, Bradford Royal Infirmary, 

Duckworth Road, 

Bradford, BD9 6RJ

PLOS

The Bradfield Centre

184 Cambridge Science Park

Milton, Cambridge CB4 0GA

United Kingdom

Dear Veincent Christian Pepito, Reviewers and Editorial Team,

We thank the reviewers for their comments and think the revisions we have made in response have strengthened this paper considerably. Our responses to the reviewers’ suggestions are given in bold after each statement. 

The page/paragraph numbers relate to the file: AEllwood_Little Minds Matter_Revised manuscript with Track Changes

I want you to clarify what specific type of "evaluation" you are carrying out. Is it an impact evaluation? Is it a process evaluation? This is to manage expectations for the reader.

This protocol describes a systems approach to evaluating the process of implementation. Therefore, this would align mostly closely with a process evaluation. The abstract and introduction now make this clearer for the reader in the project/paper aims. (P.2 L.28 and P.5 L.102)

I want you to give more details about what the intervention is by actually showing its theory of change (i.e., from inputs to process, to output, to outcome, and to impact), instead of just its parts. It would also be helpful what specific parts or aspects of the ToC will you be actually evaluating (which will answer Comment 1).

We have added clarity to the section on the intervention by including a more detailed overview of the intervention by amending Figure 1 and a description of the theoretical underpinnings of the different strands of LMM activity, and how it is expected that these strands will enact change across the system. (P.8 L. 154-159 and Figure 1)

One of the objectives is to assess whether LMM increased knowledge and understanding of IMH (Line 144-145). However, I do not actually see what specific methodology you will use to answer this objective and how you will attribute any change in knowledge to the LMM program instead of other externalities.

This objective will be answered by questioning participants of group three about their understanding of IMH, how this has changed over the duration of the project at both time points. Data collected at both times will be used to explore this change. (See added Table of objectives and data processes)

On a more general note, I would appreciate it if you could specify how you would answer each of the objectives you have listed in Lines 134-145. Who are the respondents, what is the evaluation design used, what frameworks should be used, what analysis methods should be used, etc. I think you will also be using an implementation research framework to answer the facilitators and barriers objective, I surmise?

We have developed a table which will add clarity to how the methods will address the objectives. This table replaces figure 3 and should provide more clarity on this. Systems mapping is a young but growing field of evaluation and we are contributing to this by exploring the opportunities and challenges to its implementation via conducting a systems approach in a complex existing service. (See added Table of objectives and data processes)

It is exemplary that you want to document how the embedding of the LMM affected the status quo. I want you to scale this up by also describing any assessments of unintended consequences that you would be doing on top of what has been described.

This is a valuable point, as this study is qualitative in nature the findings will draw out unintended consequences of the intervention in particular by exploring change in the system, including the displacement of resources over the duration of LMM service provision. However, we are unable to conduct any formal assessment of unintended consequences as we do not have access to financial records and we are reliant on representatives of services remembering and reporting these consequences. As with all qualitative research we are relying on services being willing to engage with interviews to disclose this information. (P.19 L.380-382)

Minor comments:

1. Do not confuse efficacy with effectiveness.

2. Fix referencing.

We have adjusted the terminology to reflect the real-world applicability of this implementation evaluation. Therefore, effectiveness and efficacy have been removed from the text throughout the paper. Referencing has been checked and amended.

The aim of the study states: “To describe the evaluation of LMM intervention using a systems evaluation approach”. Are there specific objectives to the study? In line 131, the authors have explained that it will look into how the system has changed over time and the dynamic relationship of the changes. Lines 136 onwards also lists study objectives. Clarify the general and specific objectives of the study and include these in the introduction.

For greater clarity, the distinction between the aim and objectives has been clarified rather than using the language of general and specific objectives. We have revised the submission to reflect this change throughout, but particularly in the introduction and in the objectives table. (P.5 L.104- P6. L113)

The introduction describes the significance of the infant mental health and the program. Since the main objective of the proposal is to evaluate, improve this section with additional literature on the systems-based approach to evaluation to help readers better understand the purpose of the protocol.

Detailed description of the systems approach has been made in the methods sections. Reference has now been included in the introduction to systems-based approaches to introduce this theory prior to detailed description in the methods section. (P.5 L.96-99 and P. 9 L.172-175)

Provide an illustration of the conceptual framework of the systems approach to evaluation

The most appropriate conceptual framework for the systems approach to evaluation is recommended by the Medical Research Council. Explicit reference has been made to this as a framework now. (P. 9 L.172-175)

It was difficult to follow what the research group’s methodology is for the data collection. Could this be better supported through a straightforward summary or perhaps illustrated through a diagram? Also, the differentiation of the participant groups through a table that summarizes the data to be obtained, data collection tool to be used, type of analysis, etc.

Please see the objectives table which replaces Figure 3 as described previously which should provide additional clarity on the objectives and approaches used to address them.

Outline potential assumptions and describe process more systematically to allow readers, policy-makers and other researchers to understand and even replicate the proposed approach to evaluation

Steps have been taken to make the underpinning assumptions of the intervention and the research design clearer to the reader throughout the paper by responding to other comments and feedback. The paper has been closely reviewed by two co-authors to ensure the information necessary for study replication is all contained in the paper without unnecessary duplication. We hope that the addition/revision of the figures also aides this.

Since this is a program that is already running, how long has it been running?

An addition has been made to the introduction to indicate when the LMM service provision began. (P. 4 L.75-76)

How is the service availed within the study setting?

This service can be accessed by professionals through standard processes of contact such as telephone and email. Professionals can contact the service for advice and support and also refer families into the service for therapeutic work. In addition to this families are able to self-refer to the service through a website or telephone contact. (P.8 L162-166)

What is the basis of the system map? Is there a guiding framework being followed particularly the 3 categories for eh core objectives that services remit?

A sentence has been added to show how the system map will be draw up using the team knowledge of the system and the service design documentation given the close relationship that the research team have with the project and wider system. (P.11 L216-219)

Clarify the inclusion criteria (Line 177). Line 189 describes targeting staff working as team lead or in managerial levels. This and other inclusion criteria may be most helpful in understanding the process if it were included in the eligibility list describing the participants.

This addition has been made to the inclusion criteria. (P12 L247-248)

The number of respondents for the groups are different. Describe the basis of the sampling.

The reasons for differing numbers for participant recruitment across the groups is based upon the group definitions, for instance the service design group consisted of a small number of people and LMM is provided by a set number of individuals. Conversely it is anticipated that a large number of services will interact with LMM. Therefore, a large number of participants make up this group when engaging with one member of each team. This may be seen in the example system map, figure 3. Further detail has been added to the methods section to provide clarity. (P.12 L252-260)

Since the group will not be stopping based on data saturation, how will the research team ensure that adequate information has already been gathered?

Data collection is dictated by the number of services identified in the system map, initial scoping suggests that these numbers will enable inclusion of individuals from a range of services in the early years service system. This has been clarified within the ‘participants’ section. (P.12 L252-260)

The limitation on the pre-intervention context should be supported by secondary literature review.

Additions have been made to the introduction to reflect why the intervention was considered a need within the local area and the value of systems approaches examining real world implementations of interventions. (P4. L.75-79)

Conclusion states that the study will evaluate whether a services has become embedded and influences a system. Which areas of the protocol will be answering this objective?

This is the primary aim of the study and therefore the aim will be met by the study. The objectives are now more clearly outlined within the objectives table.

Improve language and grammar of the manuscript to provide a clearer and cohesive narration of the project.

Two co-authors have extensively reviewed the manuscript to check the grammar, clarity and cohesion of the paper.

Before embedding the Little Minds Matter in early years system, its validity as an effective intervention in improving infant mental has to be established. Please provide more information on this, i.e. initial outcomes studies on improvement of parent-infant relationships, etc. after completing the program.

This evaluation is part of establishing evidence for the effectiveness of Infant Mental Health services generally as the existing evidence base is limited. Little Minds Matter has been developed specifically with the local population in mind with stakeholder input. Other Infant Mental Health services in the UK exist as described in the introduction. Additions have been made to this section regarding a gap in provision in Bradford and the design of the intervention in the intervention section. (P4. L.75-79 and P.8 154-159)

Line 169. Would it be possible to include recipients of the program as another stakeholder group as it is noted that those interviewed are all service providers

We acknowledge the value that could be added by including families in understanding the benefits of the service. It is not currently possible to interview the families in receipt of therapy for this systems evaluation and it was outside the scope of this study as we wanted to explore the understanding of professionals within the system of that system. However further evaluation may explore the experience of therapy.

Line 222. On the contrary, wouldn't it be beneficial for Group 3 to review the service map to see how LMM relates to the other services in the early years system to decrease the chances of it duplicating existing services and also to see how it can work synergistically with other programs?

Although it may be beneficial for services to understand the early years system in which they operate it is beyond the scope of this research to facilitate this. The purpose of the interviews is to understand how each service individually interacts with LMM, including any overlap in their remit and provision. Group 2 (those who provide LMM) participants will be expected to describe how their service interacts with the other services using the system map including overlap between their service and others in the wider system. This has been expanded upon the data collection section. Additionally, we also aimed to minimise the burden of participation on group 3 to maximise recruitment, therefore 30-45 minute interviews where key questions were asked rather than using the time to review the map was seen as a priority. We did not expect the participants of group 3 to find time to review the map prior to interview unlike groups 1 and 2 who had more investment in the service and evaluation. The group 1 and 2 interviews were double the length and more in-depth. (P.15 L.297-301)

- Line 295. I am not clear about what your parameters are about LMM being "embedded" and "influencing" the system

These concepts are now defined more clearly within the introduction section. (P.4-L81-92)

We hope that these changes and additions to the paper in line with your valuable feedback and comments are acceptable and await your response.

Please direct all correspondence concerning this manuscript to Alisonellwood@bthft.nhs.uk

Thank-you for your consideration of this manuscript.

Sincerely 

Alison Ellwood

On behalf of co-authors: Dr. Sarah Masefield, Dr. Josie Dickerson, Dr. Sarah Blower, Dr. Rachael Moss, Dr. Sara Ahern 

Bradford Institute for Health Research

Temple Bank House, Bradford Royal Infirmary, Duckworth Road, Bradford, BD9 6RJ

---

## [Decision Letter · Decision Letter 1]

12 Oct 2023

Study protocol for a systems evaluation of an infant mental health service: integration of ‘Little Minds Matter’ into the early years system.

PONE-D-23-15381R1

Dear Dr. Ellwood,

We’re pleased to inform you that your manuscript has been judged scientifically suitable for publication and will be formally accepted for publication once it meets all outstanding technical requirements.

Kind regards,

Veincent Christian Pepito

Academic Editor

PLOS ONE

Additional Editor Comments (optional):

Congratulations on your work! Few comments on your final draft:

1. Emphasize that this is a process evaluation (not process of evaluation) to manage expectations

2. Review capitalization, grammar, and ensure that the font size, fonts are uniform throughout the manuscript to avoid typesetting errors.

Reviewers' comments:

Reviewer's Responses to Questions

**Comments to the Author**

1. Does the manuscript provide a valid rationale for the proposed study, with clearly identified and justified research questions?

Reviewer #1: Yes

Reviewer #2: Yes

2. Is the protocol technically sound and planned in a manner that will lead to a meaningful outcome and allow testing the stated hypotheses?

Reviewer #1: Yes

Reviewer #2: Yes

3. Is the methodology feasible and described in sufficient detail to allow the work to be replicable?

Reviewer #1: Yes

Reviewer #2: Yes

4. Have the authors described where all data underlying the findings will be made available when the study is complete?

Reviewer #1: Yes

Reviewer #2: Yes

5. Is the manuscript presented in an intelligible fashion and written in standard English?

Reviewer #1: Yes

Reviewer #2: Yes

6. Review Comments to the Author

You may also provide optional suggestions and comments to authors that they might find helpful in planning their study.

Reviewer #1: Thank you for the meticulous work and considerations in revising your manuscript. It has gained a lot more clarity, and I have no further comments.

Reviewer #2: This is a much-improved protocol. It is easier to understand and more cohesive. Congratulations!

Table 1 is a welcome addition as it showed how the objectives will be collected and analyzed.

I suggest you use the active form of sentences for your final draft (ex. Line 325, etc)

Line 329- indicate how the participant will be selected from the group so as to assure non-biased selection

Line 336 - "We will not employ restrictions on coding, such as stopping analysis in response to data saturation because of the variety of professionals and services interviewed..." - please clarify what you mean by not stopping analysis when you have reached data saturation, this might mean redundant results or a lengthy discussion about similar themes.

7. PLOS authors have the option to publish the peer review history of their article (what does this mean?). If published, this will include your full peer review and any attached files.

Reviewer #1: **Yes: **Angel Belle C. Dy

Reviewer #2: **Yes: **Lourdes Bernadette Sumpaico-Tanchanco

---

## [Editor Report · Acceptance letter]

11 Dec 2023

PONE-D-23-15381R1 

Study protocol for a systems evaluation of an infant mental health service: integration of ‘Little Minds Matter’ into the early years system 

Dear Dr. Blower:

I'm pleased to inform you that your manuscript has been deemed suitable for publication in PLOS ONE. Congratulations! Your manuscript is now with our production department. 

Kind regards, 

on behalf of

Mr Veincent Christian Pepito 

Academic Editor

PLOS ONE